# Heat and dehydration induced oxidative damage and antioxidant defenses following incubator heat stress and a simulated heat wave in wild caught four-striped field mice *Rhabdomys dilectus*

**Paul J. Jacobs**[1]*, **M. K. Oosthuizen**[1], **C. Mitchell**[2], **Jonathan D. Blount**[2], **Nigel C. Bennett**[1]

**1** Department of Zoology and Entomology, Mammal Research Institute, University of Pretoria, Pretoria, South Africa, **2** Centre for Ecology and Conservation, College of Life & Environmental Sciences, University of Exeter, Penryn, Cornwall, United Kingdom

* pauljuanjacobs@gmail.com

## Abstract

Heat waves are known for their disastrous mass die-off effects due to dehydration and cell damage, but little is known about the non-lethal consequences of surviving severe heat exposure. Severe heat exposure can cause oxidative stress which can have negative consequences on animal cognition, reproduction and life expectancy. We investigated the current oxidative stress experienced by a mesic mouse species, the four striped field mouse, *Rhabdomys dilectus* through a heat wave simulation with *ad lib* water and a more severe temperature exposure with minimal water. Wild four striped field mice were caught between 2017 and 2019. We predicted that wild four striped field mice in the heat wave simulation would show less susceptibility to oxidative stress as compared to a more severe heat stress which is likely to occur in the future. Oxidative stress was determined in the liver, kidney and brain using malondialdehyde (MDA) and protein carbonyl (PC) as markers for oxidative damage, and superoxide dismutase (SOD) and total antioxidant capacity (TAC) as markers of antioxidant defense. Incubator heat stress was brought about by increasing the body temperatures of animals to 39–40.8°C for 6 hours. A heat wave (one hot day, followed by a 3-day heatwave) was simulated by using temperature cycle that wild four striped field mice would experience in their local habitat (determined through weather station data using temperature and humidity), with maximal ambient temperature of 39°C. The liver and kidney demonstrated no changes in the simulated heat wave, but the liver had significantly higher SOD activity and the kidney had significantly higher lipid peroxidation in the incubator experiment. Dehydration significantly contributed to the increase of these markers, as is evident from the decrease in body mass after the experiment. The brain only showed significantly higher lipid peroxidation following the simulated heat wave with no significant changes following the incubator experiment. The significant increase in lipid peroxidation was not correlated to body mass after the experiment. The magnitude and duration of heat stress, in conjunction with dehydration, played a critical role in the oxidative stress experienced by

**Data Availability Statement:** All relevant data are within the manuscript and its Supporting information files.

**Funding:** This research was supported by a DST-NRF SARChI research chair for Mammal Behavioural Ecology and Physiology to NCB and a University of Pretoria doctoral research bursary and a University of Pretoria department of research and innovation international cooperation postgraduate exchange bursary to PJJ. The funders had no role in study design, data collection and analysis, decision to publish, or preparation of the manuscript.

**Competing interests:** The authors have declared that no competing interests exist.

each tissue, with the results demonstrating the importance of measuring multiple tissues to determine the physiological state of an animal. Current heat waves in this species have the potential of causing oxidative stress in the brain with future heat waves to possibly stress the kidney and liver depending on the hydration state of animals.

## Introduction

Extreme temperature climatic events (heat waves) are a real threat to animal biodiversity through a variety of lethal and sublethal effects [1–4]. Lethal heat stress from heat waves are likely due to dehydration and cellular heat damage [5], with just a single day of extreme temperatures leading to a mass die-off of an endangered bird the Carnaby's Cockatoo (*Calyptorhynchus latirostris*) [6]. Several other mass die-off events have occurred in the last 20 years resulting in the deaths of humans, bats and birds [4, 7–9]. Sublethal effects of repeated exposure to extreme heat events may include loss of body condition, compromised reproduction and reduced cognitive performance, which can result in overall population declines [3]. These heat waves are predicted to become more frequent and intense in the Anthropocene [8, 10–12], highlighting concerns for species extinctions [13].

Small animals are generally assumed to circumvent the effects of climate change due to the use of microsites within a habitat to escape extreme temperatures [14–17]. In addition to the use of microsites, smaller animals have a larger surface area to volume ratio allowing for rapid heat loss assuming air temperature is below skin temperature [18]. A larger surface area to volume ratio can also be detrimental since rapid heat loss is accompanied by rapid heat gain and without the presence of microsites may drastically compromise small animal survival [14].

Animals can behaviourally alleviate the effects of heat stress, through drinking more water [19–21], reduce thermogenic activity by eating less [22, 23] and lowering activity rate [24, 25]. Rodents do not sweat (except from their footpads) [26, 27], or pant to increase evaporative water loss [28]. Instead, rodents primarily use saliva spreading for evaporative heat loss, while rodent species that do not utilise saliva spreading suffer exaggerated responses to heat stress [20].

One pattern observed as a consequence of anthropogenic climate change, is the geographic shift of species to escape elevated temperatures [10]. One southern African species, the mesic four-striped field mouse, *Rhabdomys dilectus* (de Winton, 1987), has been proposed to be at risk and may undergo a geographic shift to escape climate change [29, 30]. The study species prefers the grassland and savannah biomes with ground cover and water [19], with the ground cover providing a thermal buffer to avoid extreme temperatures [29]. The congeneric desert living four striped field mouse *Rhabdomys pumilio* (Sparrman 1784) has a thermoneutral zone (TNZ) of 32˚C [31], with the mesic species' TNZ still to be determined.

Aerobic organisms constantly produce reactive oxygen species (ROS) from metabolism, and utilise antioxidants to reduce excessive ROS in order to maintain redox balance [32, 33]. Despite the negative connotation to ROS, ROS are important to cellular signalling [34], to inflammation response [35], altering glucose uptake and metabolism [36], immune response [37], allowing for the preparation to deal with hypoxic stress [38] and osmoprotective signalling [39].

Heat stress and dehydration independently can disrupt this balance [40–42]. Heat stress can disrupt this balance through excessive metabolic ROS production and lowered antioxidant activity, resulting in a state of oxidative stress [43, 44]. Dehydration disrupts this balance

caused by hyperosmolality induced from cellular shrinkage and compromised membrane functionality [45–47]. This excessive oxidative imbalance in favour of ROS can ultimately lead to oxidative damage to DNA [48, 49], lipids [50] and proteins [51, 52]. Oxidative stress can reduce cognitive and motor performance [53], fertility [54, 55] and life expectancy [56, 57]. Therefore, biomarkers of oxidative stress can provide highly relevant insights into the physiological state of an organism [58, 59], making it possible to establish whether an animal is vulnerable to changes in its environment. However, we are not aware of any previous studies that have measured oxidative stress levels in a variety of tissues as a consequence of exposure to different thermal regimes in the four striped field mouse.

We measured the heat stress response in terms of oxidative damage and antioxidant defense of four-striped field mouse following an incubator heat stress and dehydration experiment and a simulated heat wave. The incubator heat stress is a whole-body fever range exposure, which is a representation of an animal's upper limit and failing thermoregulatory system. The simulated heat wave represents current heat wave extremes where the animals were caught (collected and simulated weather station data). Oxidative damage and antioxidant defense were measured in three organs, namely the liver, kidney and brain. These three tissues account for over 60% of the body's resting metabolic rate, at least in humans [60, 61]. Brain was chosen for its susceptibility to thermal stress, and in particular its involvement in multiple organ dysfunction during heat stress [62, 63]. The kidney was selected based on its importance in water retention and an abundance of long-chain polyunsaturated fatty acids in the composition of renal lipids [64], such molecules being especially susceptible to lipid peroxidation [65]. Lastly, liver was chosen because it is an important source of glutathione [66], the most important antioxidant in determining total antioxidant capacity (TAC) in tissues [67]. In addition to the liver's relevance to TAC, heat stress is also associated with elevated oxidative damage and antioxidant defense in this tissue [68, 69], due in part to fluctuations in labile iron [70].

We predicted that the four-striped field mouse would demonstrate increased susceptibility to heat stress induced oxidative stress in the incubator experiment as compared to the simulated heat wave relative to their respective controls. In this context we define oxidative stress as significant changes in oxidative damage and antioxidant defense compared to the controls.

Oxidative damage and antioxidant defense occur concomitantly, therefor increases in antioxidant defense (superoxide dismutase (SOD) enzyme or TAC) will demonstrate increased oxidative stress, whereas oxidative damage (lipid and/or proteins) will represent a compromised antioxidant defense.

## Methods

Animal Ethics Committee University of Pretoria (AEC) with approval number EC008-17.

### Animal maintenance

Ten adult male four-striped field mice, were wild caught and used in the incubator heat stress experiment and thirteen adult male animals were wild caught for the simulated heatwave. Males were used in order to prevent sex differences in oxidative stress [71–73]. The field mice were captured at the Rietvlei nature reserve (3800ha, Centurion, South Africa, -25° 53' 29.39" S, 28° 17' 22.80" E), using metal Sherman traps (26 cm x 9 cm x 9 cm), baited with a mixture of oats and peanut butter. Rietvlei is a local nature reserve and we obtained written permission from the manager to perform work here. Mice used for the incubator heat stress experiment were caught between March 2017-June 2017; whereas those for the simulated heat wave experiment were captured between January 2019-March 2019. Once caught, mice were maintained in field cages and subsequently transported to the Zoology and Entomology Department, at

the University of Pretoria. The mean body mass before the experiment of the striped field mouse used for the incubator heat stress experiment was 62.7 ± 16.4 standard deviation (SD) g, whereas the mean body mass before the experiment for the simulated heat wave experiment was 43.1 ± 5.6 (SD) g. The mice were weighed weekly to assess body condition. Mice were weighed to ± 0.1 g prior to and after an experiment. Animals were housed in a room at the University of Pretoria and acclimated to a 12L:12D photoperiod, 40% RH and temperature of around 23°C. This temperature also resembles the average temperature experienced during a summer day. Mice were maintained in captivity for at least 60 days prior to their use for both experiments. This initial acclimation period was performed to minimise the influence of stress by bringing wild animals into captivity. All mice were housed individually in 40 x 25 x 12 cm standard laboratory mouse containers lined with wood shavings and a rock, toilet rolls, a small plastic container for a nest, and with tissue paper for nesting material. Animals were provided *ad libitum* with water and food. Food was provided every second day in the form of sunflower seeds, corn, banana, carrot, apple shavings or sweet potato slices. The cages were cleaned weekly.

## Experiment 1: Incubator heat stress experimental design and protocol

The heat stress treatment was used to investigate the oxidative markers following a 6-hour whole-body fever range hyperthermia acute heat stress (39–40.8°C body temperature ($T_b$)) and dehydration stress in an incubator. This incubator heat stress protocol followed that of Ostberg, Kaplan [74], a period of 6-hour of $T_b$ around 39–40.8°C $T_b$ was used to induce whole-body fever range hyperthermia that could result in oxidative heat stress response.

A control group maintained at an ambient temperature ($T_a$) of 25°C (animal core $T_b$ averaged ±36.6°C) served to determine oxidative markers without the influence of heat stress. Individuals were randomly allocated to the heat-stressed group and the control group. Lastly, food and water were not provided during this experiment to prevent metabolic and humidity increases respectively, which will influence heat loss.

In order to determine animal core $T_b$, temperature-sensitive PIT tags (BioTherm, Identipet), which can be read with a PIT tag reader, were injected intraperitoneally using sterile syringes at least a week prior to the experiment. Due to the length of the experiment, all individuals were injected with 1ml of saline on day 1 of the experiment just prior to the thermal manipulation to help animals buffer against dehydration over the 6-hour time period.

The incubator was pre-heated to 41°C $T_a$ (heat stress) or 25°C $T_a$ (control) prior to the animal being placed inside the incubator. Animals were subsequently transferred to an experimental chamber of 25 x 13 x 16 cm, and placed in the incubator. Lights were switched off inside the incubator to minimise extra heat from light sources inside the incubator. Animal core $T_b$ was monitored using a PIT reader antenna; incubator temperature ($T_a$) was modulated as required to maintain $T_b$ at around 39–40.8°C, the required range (S1 Table). Readers were set to record every 10 seconds. After 6-hours, animals were removed from the experiment.

## Experiment 2: Simulated heat wave experimental design and protocol

**Control, transition and heat wave simulation.**   The four striped field mice were transferred to large plastic individual containers (60 x 40 x 30 cm) lined with wood shavings, a small plastic container for a nest and a toilet roll and tissue paper provided as nesting material. Lighting was set to a 14L:10D long day schedule, which included 4 hours of 'twilight' with increasing and decreasing light intensities simulating dawn and dusk respectively. The long day photoperiod was accompanied by a typical temperature cycle, which was determined through climatic data obtained from the South African weather service (Fig 1).

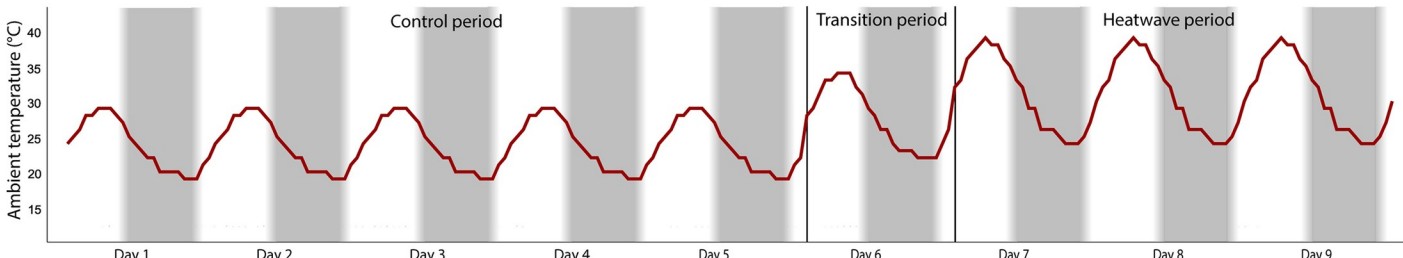

**Fig 1. Photoperiod and temperature profile used to simulate control, transition, and heat wave temperatures.** The white areas represent day time and shaded areas represent night time. The black lines represent changes in experimental condition, with the control temperatures lasted 5 days, transition temperatures lasted 1 day and the simulated heat wave lasted 3 days. The red line represents the temperature cycle that animals were exposed to during the 9-day experiment.

The control group of mice were set up to determine the influence of a daily temperature cycle from an average summer day (calculated from the South African Weather service data) to simulate minimal to no heat stress at least when compared to a heat wave. Control mice were maintained for nine days on a cycle that oscillated between a minimum of 19˚C and a maximum of 29˚C (Fig 1). In contrast, for the heat wave simulation, all animals were first maintained at control temperatures for five days, then transitioned to temperatures that oscillated between a minimum of 22˚C and a maximum of 34˚C for one day, followed by heat wave temperatures which oscillated between a minimum of 24˚C and a maximum of 39˚C for a period of full three days. During changes in the temperature (control to transition, and transition to heat wave), temperatures were set to represent the new treatment condition starting at 9:00 am. During experimentation, the experimenter influence was kept to a minimum, with a 30 min interval (between 8:00 am-8:30 am) for feeding and checking the general welfare of the animals. The mice were given a fixed amount of water and food (sunflower seeds every day, fresh fruit/vegetables every second day).

## Calculation of simulated heat wave temperatures

Weather station data provided by the South African Weather Service were corrected using a temperature-humidity index (THI). This index was calculated from the wet and dry bulb air temperatures for a particular day according to the following formula: THI = 0.72 (W + D) + 40.6, where W is wet bulb and D is dry bulb temperature in degree centigrade. The following website was used to convert climatic data values to heat index values (http://www.wpc. ncep.noaa.gov/html/heatindex.shtml). A THI was used to correct for constant humidity inside the temperature control rooms such that animals would feel the perceived air temperatures as actual temperature conditions, instead of elevated temperatures with higher humidities.

## Euthanasia and tissue excision

All mice were euthanised with an overdose of isoflurane immediately at the end of each respective experiment. All samples were collected at the same time to prevent daily rhythm effects, with tissues collected in the same order within 10 minutes at post-mortem with an approximate 1-minute interval between tissues. This was done to prevent and/or minimise proteins and metabolites from denaturing from dissection to being flash frozen. The liver, kidney, and brain were collected and flash-frozen in liquid nitrogen, and subsequently stored at -80˚C until analysis (less than 6 months for all tissues).

## Analyses of oxidative damage and antioxidant defense

**Tissue homogenization.** Tissues (liver, brain and kidney) were homogenised on ice by 10% weight per volume in 20 mM HEPES (N-2 hydroxyethylpiperazine-N9-2-ethanesulfonic acid) buffer on an Ultra Turrax T18 Basic Homogenizer (IKA, Staufen, Germany) for the incubator heat stress and on an Ultra Turrax T25 Basic Homogenizer (IKA Labortechnik, Germany) for the simulated heat wave experiment. Homogenates were then stored in a -80°C freezer until the time of analysis (less than 6 months for all tissues).

**Malondialdehyde: Incubator heat stress and simulated heatwave.** The concentrations of MDA in all tissue homogenates (i.e. liver, kidney and brain) were measured by high-performance liquid chromatography (HPLC) using standard techniques [75]. Prepared samples were injected 20 μL into an Agilent HPLC system (InfinityLab Solutions, California, USA) fitted with a 5 μm ODS guard column and a Hewlett-Packard Hypersil 5 μ ODS 100 x 64.6 mm column maintained at 37°C. The mobile phase was methanol-buffer (40:60, v/v; 50 mM anhydrous solution of potassium monobasic phosphate at pH 6.8), running isocratically over 3.5 min at a flow rate of 1 ml per min. Data were collected using a fluorescence detector (RF2000; Dionex) set at 515 nm (excitation) and 553 nm (emission). For calibration, a standard curve was established using a TEP stock solution (5 μM in 40% ethanol) serially diluted using 40% ethanol. Results are expressed as μMol MDA per g homogenate.

**Protein carbonyl: Incubator heat stress.** PC concentrations were measured from tissue homogenates (i.e. liver, kidney, and brain). Oxidation or oxidative cleavage of proteins results in the production of carbonyl groups following standard technique [76], which covalently react with 2,4-dinitrophenylhydrazine (DNPH) to form 2,4-dinitrophenyl (DNP) hydrazone. DNP is detected via spectrophotometry at a wavelength of 370 nm [77]. Our study protocol differed by using 1ml of 20% TCA instead of 125μL of 50% TCA. Absorbances were read using a Spectramax M2 plate reader (Molecular Devices Corp., Sunnyvale, CA, USA). Samples were run in duplicate with a repeatability of $r = 0.99$ between control and samples. Protein content was determined using the Bradford assay using a bovine serum albumin (BSA) standard curve. 180 μL of guanidine-HCL solution was added to 20 μL of control sample (HCL solution) in a 1:10 ratio. Absorbances were read at 280nM using a Spectramax M2 plate reader (Molecular Devices Corp., Sunnyvale, CA, USA). Samples for the Bradford assay were run in duplicate with a repeatability of $r = 0.99$. The results are expressed in μMol per g protein.

**Protein carbonyl: Simulated heat wave.** PC was measured from the tissue homogenates (i.e. liver, kidney, and brain). PC concentrations were measured using a commercially available kit (Sigma-Aldrich, cat. no. MAK094, MO, USA), reading the absorbance of samples using an Eon high-performance microplate spectrophotometer (BioTek Instruments Inc., USA). Protein content of each sample was analysed using a BCA assay (Sigma-Aldrich, cat. no. BCA1 and B9643, MO, USA) using a BSA standard (Sigma-Aldrich, cat. no. P0914, MO, USA). PC samples were run in duplicate with a repeatability of $r = 0.70$. BCA samples were run in duplicate with a repeatability of $r = 0.83$. Results are expressed as μMol per g protein.

**Superoxide dismutase: Incubator heat stress and simulated heatwave.** SOD activity was measured in all tissue homogenates (i.e. liver, kidney and brain). SOD is an enzymatic antioxidant that catalyses the dismutation of superoxide anions to oxygen and hydrogen peroxide [78]. Analyses were performed following standard techniques [79]. SOD content was measured with a commercially available kit (Superoxide Dismutase Assay Kit, Cayman Chemical Co., Ann Arbor, MI, USA) that measures the percentage of superoxide radicals that undergo dismutation in a given sample. Absorbance was read at 450 nm using a Spectramax M2 plate reader (Molecular Devices Corp., Sunnyvale, CA, USA). Samples were run in duplicate with a repeatability of $r = 0.82$. The results are expressed in units of SOD activity per g homogenate.

**Total antioxidant capacity: Incubator heat stress.** TAC in homogenates of liver, kidney and brain were quantified using a commercially available kit (Antioxidant Assay Kit, Cayman Chemical Co., Ann Arbor, MI, USA) which measures the oxidation of ABTS (2,29-Azino-di-[3-ethybenzthiazoline sulphonate]) by metmyoglobin, which is inhibited by non-enzymatic antioxidants contained in the sample. Oxidized ABTS is measured by spectrophotometry at a wavelength of 750 nm. The capacity of antioxidants in the sample to inhibit oxidation of ABTS is compared with the capacity of known concentrations of Trolox, and the results are expressed as mM of Trolox equivalents per g homogenate. Samples were run in duplicate with a repeatability of $r = 0.90$.

**Total antioxidant capacity: Simulated heat wave.** TAC in homogenates of liver, kidney, and brain were quantified using a commercially available kit (Sigma-Aldrich, cat. no. MAK187 and D2650, MO, USA), following standard techniques [80]. The concentration of large and small molecular antioxidants and total antioxidant capacity can be measured through the conversion of $Cu^{2+}$ ions to $Cu^+$, with the reduced $Cu^+$ ion chelated with a colourimetric probe read at an absorbance of 570nm. The TAC is compared to an antioxidant activity standard in Trolox equivalents (in 4-20nmole/well). Absorbances were read using an Eon high-performance microplate spectrophotometer (BioTek Instruments Inc., USA). Samples were run in duplicate with a repeatability of $r = 0.95$. Results are expressed as mM of Trolox equivalents per g homogenate.

## Statistical analyses

One control kidney sample was lost for *R. dilectus* during the process of analyses for the simulated heatwave experiment. Data were examined for normality and outliers, where outliers were kept to maintain sample size. Normality was tested using the Shapiro-Wilk and Kolmogorov-Smirnov test. Homogeneity of variance was tested using Levene's test and Brown-Forsythe test. Data were log-transformed where normality was not observed, and we used the appropriate statistical test for unequal variances when homogeneity of variances was not observed. MDA, PC, SOD and TAC levels consisted of data with one independent variable (treatment) separated into two groups (control vs heat stressed). Each tissue was analysed separately. From this, independent samples t-tests were performed to determine the difference in the incubator heat stress and the simulated heat wave treatment from their respective controls. A repeated measures ANOVA was used to determine whether body mass (before and after) significantly changed for each treatment (control and stressed) for each experiment separately (incubator and simulated heat wave). The interactive term body mass x treatment is reported. For significant treatment effects, a partial correlation was performed to determine whether dehydration (determined through changes in body mass before and after an experiment) was significantly correlated to the oxidative marker. Significance was calculated at P<0.05. All analyses were executed using SPSS (version 26) (IBM Corp. Armonk, NY). The results are reported as means ± s.e.

## Results

### Experiment 1: Incubator heat and dehydration stress

Lipid peroxidation following the incubator heat stress experiment did not differ significantly from the control in the liver (t-test, $t_8 = 0.85$, $p = 0.42$ or brain (t-test, $t_8 = 2.01$, $p = 0.10$), but was significantly higher in the kidney (t-test, $t_8 = 2.70$, $p = 0.027$) (Fig 2A) compared to the control. Tissues did not significantly differ for protein oxidation following the incubator heat stress experiment when compared with the control (liver: t-test, $t_8 = 0.53$, $p = 0.61$; kidney: t-test, $t_8 = 1.03$, $p = 0.33$; brain: t-test, $t_8 = 1.13$, $p = 0.29$) (Fig 2B). SOD following the incubator

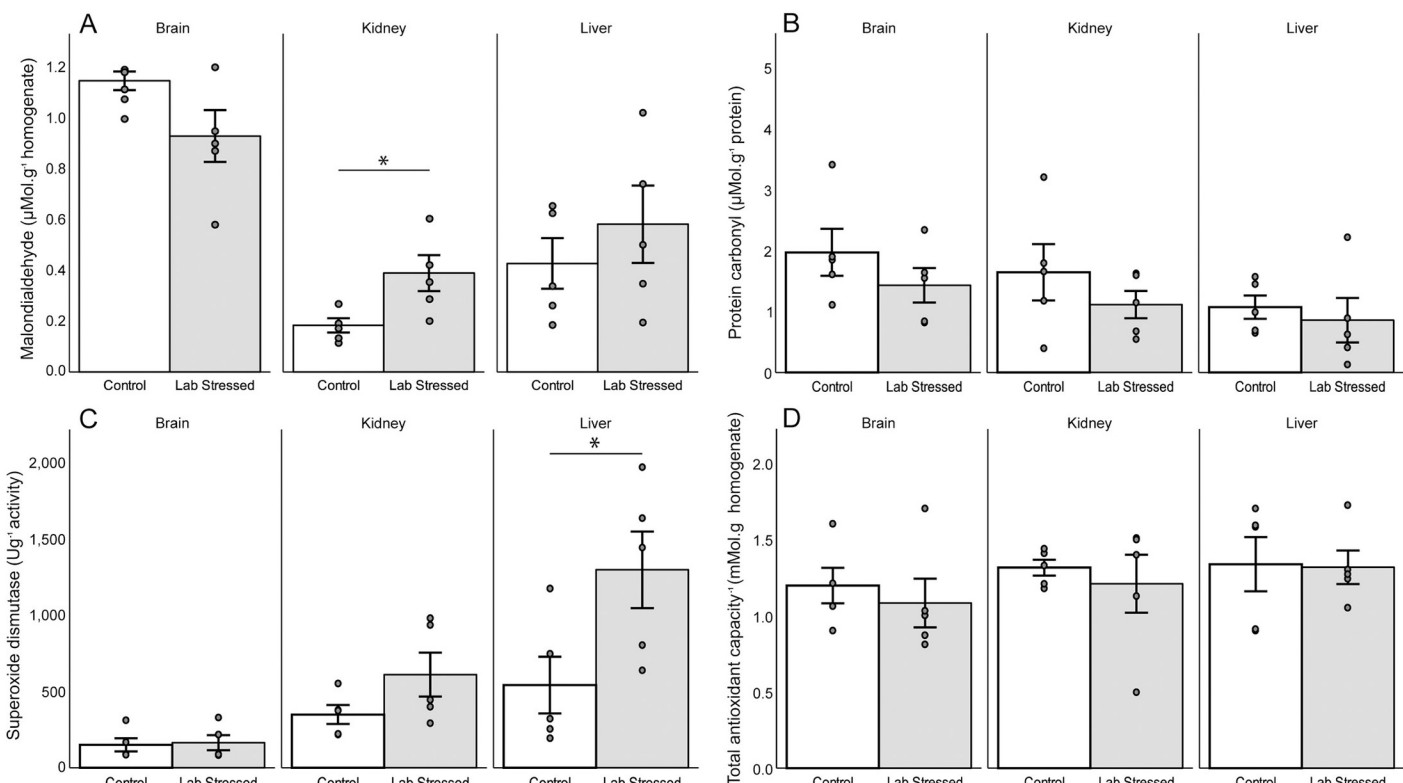

**Fig 2. The mean A) malondialdehyde B) protein carbonyl C) superoxide dismutase D) total antioxidant capacity of the brain, kidney and liver in** *Rhabdomys dilectus* **(N = 5) as a function of an incubator heat stress treatment.** Error bars represent ± s.e. Significance at $p < 0.05$.

heat stress experiment did not significantly differ from the control for the kidney (t-test, $t_8$ = 1.69, p = 0.13) or brain (t-test, $t_8$ = 0.17, p = 0.87), but was significantly higher in the liver (t-test, $t_{7.34}$ = 2.42, p = 0.045) compared to the control (Fig 2C). TAC following the incubator heat stress experiment did not differ significantly in any tissue (liver: t-test, $t_8$ = 0.94, p = 0.93; kidney: t-test, $t_8$ = 0.72, p = 0.49; brain: t-test, $t_8$ = 0.58, p = 0.58) (Fig 2D). The incubator heat stress group had significantly greater change in body mass4.22 ± 1.79%, compared to the control group, which had 0.45 ± 0.66% change in body mass after the experiment ($F_1$ = 23.47, p = 0.001) (S2 Table). Liver SOD activity (N = -0.753, df = 7, p = 0.019) and kidney MDA (r = -0.763, df = 7, p = 0.017) both had a significant negative correlation to body mass after the experiment when controlling for body mass before the experiment (S2 Table).

## Experiment 2: Simulated heat wave

Following the simulated heat wave, lipid peroxidation did not significantly differ from the control for the liver (t-test, $t_{11}$ = 0.78, p = 0.45) or kidney (t-test, $t_{10}$ = 0.35, p = 0.74), but was significantly higher for the brain (t-test, $t_{11}$ = 3.10, p = 0.010) (Fig 3A). Protein oxidation following the simulated heat wave did not significantly differ from the control for any of the three tissues (liver: t-test, $t_{11}$ = 1.63, p = 0.13; kidney: t-test, $t_{10}$ = 1.67, p = 0.13; brain: t-test, $t_{11}$ = 1.74, p = 0.11) (Fig 3B). Following the simulated heat wave, SOD did not significantly differ from the control for all tissues (liver: t-test, $t_{11}$ = 0.38, p = 0.71; kidney: t-test, $t_{10}$ = 0.35, p = 0.73; brain: t-test, $t_{11}$ = 0.14, p = 0.89) (Fig 3C). Tissues did not significantly differ from the control in TAC following the simulated heat wave (liver: t-test, $t_{11}$ = 2.18, p = 0.052; kidney: t-

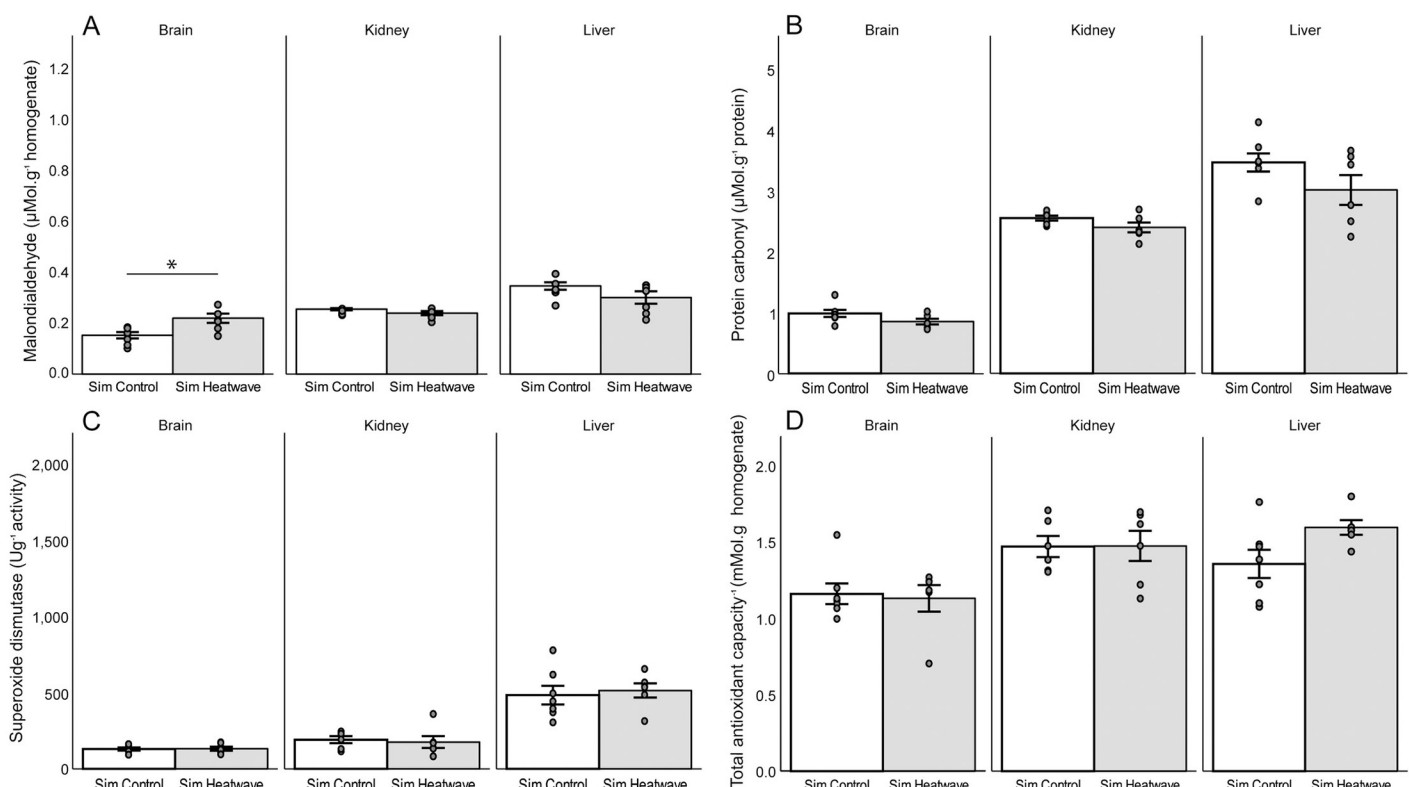

**Fig 3. The mean A) malondialdehyde B) protein carbonyl C) superoxide dismutase D) total antioxidant capacity of the brain, kidney and liver *Rhabdomys dilectus* as a function of a simulated heat wave.** Control tissues N = 7 (kidney N = 6) and simulated heat wave N = 6. Error bars represent ± s.e. Significance at p<0.05.

test, $t_{10}$ = 0.038, p = 0.97; brain: t-test, $t_{11}$ = 0.27, p = 0.79) (Fig 3D). Both experimental groups had a net positive body mass with no significant difference (F1 = 2.23, p = 0.15) after the experiment, with the control group with 7.96 ± 7.36% change in body mass and the heat stress group had 1.12 ± 7.28% change in body mass after the experiment (S2 Table). No significant partial correlation was observed for brain MDA (p = -0.491, df = 10, p = 0.11) (S2 Table).

## Discussion

Markers of oxidative stress significantly changed in response to two different thermal stress regimes compared to their controls, suggesting thermal and dehydration stress can alter oxidative balance in specific tissues in four striped field mice. Dehydration was determined through the change in body mass, where a body mass exceeding 2% is expected to result in a dehydrated state [81, 82]. The mean from individuals far exceeded this value and it is therefore likely that individuals were dehydrated. In the incubator control group, the mice as a whole demonstrated a net gain in body mass suggesting they were hydrated post-experiment. Body mass measurements were taken prior to the saline injection, which explained the net gain in body mass in the control group while the experimental group lost the weight of the injection and more, resulting in a net body mass loss. Following the incubator heat stress the liver demonstrated antioxidant defense through higher SOD activity preventing significant oxidative damage. In contrast to this, the kidney was susceptible to lipid peroxidation with significantly higher levels of MDA. Interestingly, the liver and kidney did not exhibit any significant

changes in oxidative damage or antioxidant defense following the simulated heat wave, with the brain exhibiting significantly higher lipid peroxidation levels.

The magnitude and duration of heat stress strongly affects how tissues will respond [83, 84]. The incubator experiment had a high magnitude and duration of heat stress, which would result in rapid heat gain and high heat loads in tissues [83]. In contrast, in the simulated heat wave, despite a much longer duration to heat stress (3 days), the overall magnitude of heat stress experienced by an animal each day was overall less (±3 hours a day). The incubator heat stress resulted in a net water loss in individuals causing dehydration, whereas sufficient water (food and drinking) during the simulated heat wave allowed individuals to hydrate themselves, possibly resulting in a reduced oxidative stress response due to no net loss of water. Thus susceptibility to heat stress-induced oxidative damage may be very dependent on the duration and magnitude of heat stress experienced, in a tissue-specific manner, as well as highlighting the importance of water availability to circumvent severe dehydration. Heat stress and dehydration together likely results in a compounding effect, and becomes exacerbated depending on the duration and magnitude of the heat stress experienced. This study demonstrated that preventing severe dehydration can minimise the effects of oxidative stress. Previously, dehydration stress has been found to not result in complete recovery after rehydration [41] and future studies may allow for the investigation of oxidative balance in tissues under similar conditions with a recovery period.

Several studies investigating heat induced oxidative stress of the liver have demonstrated elevated lipid peroxidation and reduced SOD activity [68, 85–88]. Heat load (amount of heat absorbed) during heat stress is particularly important in the liver, as the liver has shown a 21 fold increase in heat shock proteins (HSPs) (protective proteins in response to thermal stress) compared to a 12 fold increase in HSPs when the heat load was less [83]. In addition to a higher amount of HSPs produced during high heat loads in the liver, higher heat stress and/or dehydration status have resulted in the rapid upregulation of SOD activity [87, 89–91]. The liver is exposed to hyperosmotic fluids under non-pathogenic conditions and can become hyperosmotic under pathological conditions, which can result in oxidative stress [45, 92, 93]. The present study demonstrated a significant upregulation of SOD activity under heat and dehydration stress, with a finding similar in *Xenopus laevis* (Daudin 1802) where SOD activity increased [94]. Due to SOD activity being significantly increased and not decreased dehydration may have played a larger role in the oxidative stress experienced in the liver. In contrast, since no significant upregulation of antioxidant enzymes were apparent during the simulated heat wave, we believe the heat stress, along with the absence of severe dehydration, was not sufficient to cause any significant oxidative stress in this tissue.

The effect of oxidative damage and antioxidant defense in the kidney in response to disease (e.g. diabetes mellitus) [95, 96], hyperosmolar conditions [45, 97, 98] and HSPs production in response to heat stress has been well documented [83, 99–101]. However, the literature on heat stress influences by itself on the oxidative balance of kidneys is scarce; acute heat stress from exercise incurred no significant oxidative damage in Sprague-Dawley rats [102], but mice exposed to heat stress had higher lipid peroxidation and reduced SOD activity [103]. In broiler chickens, acute heat stress caused a minor decease in SOD activity along with a lower level of lipid peroxidation compared to the control [88]. Goldfish, *Carassius auratus* (Linneaus, 1758) in response to heat stress demonstrated elevated lipid peroxidation, minimal changes in SOD activity, but increased in glutathione enzymes, which demonstrated higher expression of other antioxidant enzymes in response to heat stress. Kidneys are sensitive to oxidative damage [104, 105], which is caused by hyperosmolality, which is more likely under the duress of increased temperatures resulting in dehydration [98]. In this study, the kidneys likely became heat stressed and dehydrated (observed from the % body mass deficit), which may have resulted in hyperosmolality

[98, 106]. Hypersomolality in the kidneys activate the polyol-fructokinase pathway and possibly the chronic effects of vasopressin to induce tubular and glomerular injury, both which cause oxidative damage [98]. The importance of kidney hydrative state is emphasized in this study, as the limited water availability, along with dehydration following the incubator experiment was associated with significantly increased oxidative stress. This may also explain why animals may have become active during a simulated heat wave to drink water to prevent dehydration [19], and in turn would prevent kidney oxidative stress. Overall, kidney oxidative stress may be very reliant on the urinary concentrating ability of the animals resulting in different tolerances to hyperosmolality, with desert animals better equipped with kidney oxidative stress [107–109].

The brain is known for its susceptibility to heat stroke resulting in multiple organ dysfunction [63, 110, 111]. The brain in laboratory mice undergoing acute heat stress from exercise demonstrated decreases in MDA and PC [102], however, during acute heat stress at extreme temperatures, brain SOD activity decreased accompanied by lipid peroxidation increases [62, 111, 112]. In contrast, the simulated heat wave demonstrated increased lipid peroxidation, but no changes in antioxidant enzyme activity [62, 111, 112]. Despite the magnitude and duration of heat stress in the incubator experiment, the brain did not significantly increase in oxidative damage. The duration of heat stress following the simulated heat wave was much longer and may have compromised the permeability of the blood-brain barrier (BBB) and resulted in a cascading effect of increased oxidative damage [113, 114]. Due to dehydration having a minimal effect following the simulated heat wave, it was likely that heat stress alone was sufficient to cause oxidative stress in the brain over long time periods, with more severe temperature exposures likely to be more deleterious [115].

In light of climate change, currently for a mesic crepuscular rodent the four striped field mouse, a three-day heat wave with freely available water is enough to induce oxidative damage in the brain, when measured in the absence of behavioural and physiological thermoregulation (e.g. microsites). In addition to current conditions (assuming no behavioural and physiological thermoregulation), whole-body fever range temperatures and dehydration may likely result in kidney and liver oxidative stress. The kidney will suffer oxidative damage when dehydrated and highlights the importance of water to animals during a heat stress to offset dehydration and to potentially rehydrate [19]. It is, however, uncertain to what extent oxidative damage repair mechanisms may reduce potential sub-lethal consequences from heat induced oxidative stress [41, 116–120]. Differences observed between the tissues demonstrates the importance of assessing oxidative damage and antioxidant defenses in different tissues to obtain an overview of an organism's physiological state [61].

## Supporting information

**S1 Table. The control and stressed PIT animal body temperature recordings inside the incubator throughout the 6 hour period.**
(XLS)

**S2 Table. The body mass before, body mass after and % body mass change of each individual across all experiments.**
(XLSX)

## Acknowledgments

We want to thank Prof. Chris Weldon and Prof. Duncan Cromarty for the provisioning of equipment. Ambaj Sharma for his assistance during laboratory work. We thank Ezemvelo and Rietvlei nature reserves for the cooperation for conducting research on their reserves.

## Author Contributions

**Conceptualization:** Paul J. Jacobs, M. K. Oosthuizen, Jonathan D. Blount, Nigel C. Bennett.

**Formal analysis:** Paul J. Jacobs, M. K. Oosthuizen, C. Mitchell.

**Funding acquisition:** M. K. Oosthuizen, Nigel C. Bennett.

**Investigation:** Paul J. Jacobs.

**Methodology:** Paul J. Jacobs.

**Resources:** M. K. Oosthuizen, C. Mitchell, Jonathan D. Blount, Nigel C. Bennett.

**Supervision:** M. K. Oosthuizen, Nigel C. Bennett.

**Writing – original draft:** Paul J. Jacobs.

**Writing – review & editing:** Paul J. Jacobs, M. K. Oosthuizen, Jonathan D. Blount, Nigel C. Bennett.

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
