## [Decision Letter · Decision Letter 0]

22 Jul 2020

PONE-D-20-14410

Heat induced oxidative damage and antioxidant defenses following incubator heat stress and a simulated heat wave in wild caught four-striped field mice Rhabdomys dilectus

PLOS ONE

Dear Dr. Jacobs,

Thank you for submitting your manuscript to PLOS ONE. After careful consideration, we feel that it has merit but does not fully meet PLOS ONE’s publication criteria as it currently stands. Therefore, we invite you to submit a revised version of the manuscript that addresses the points raised during the review process.

Please follow closely the recommendations of referees 1 and 2.

Please submit your revised manuscript within 60 days. If you will need more time than this to complete your revisions, please reply to this message or contact the journal office at plosone@plos.org. Please include the following items when submitting your revised manuscript:

We look forward to receiving your revised manuscript.

Kind regards,

Marcelo Hermes-Lima, PhD

Academic Editor

PLOS ONE

Journal Requirements:

3. Your ethics statement must appear in the Methods section of your manuscript. If your ethics statement is written in any section besides the Methods, please move it to the Methods section and delete it from any other section. Please also ensure that your ethics statement is included in your manuscript, as the ethics section of your online submission will not be published alongside your manuscript.

4. Please upload a new copies of Figures 1, 2 and 3 as the details are not clear. Please follow the link for more information: https://blogs.plos.org/plos/2019/06/looking-good-tips-for-creating-your-plos-figures-graphics/

Reviewers' comments:

Reviewer's Responses to Questions

**Comments to the Author**

1. Is the manuscript technically sound, and do the data support the conclusions?

Reviewer #1: Partly

Reviewer #2: No

Reviewer #3: Yes

2. Has the statistical analysis been performed appropriately and rigorously? 

Reviewer #1: No

Reviewer #2: Yes

Reviewer #3: Yes

3. Have the authors made all data underlying the findings in their manuscript fully available?

Reviewer #1: No

Reviewer #2: Yes

Reviewer #3: Yes

4. Is the manuscript presented in an intelligible fashion and written in standard English?

Reviewer #1: Yes

Reviewer #2: Yes

Reviewer #3: Yes

5. Review Comments to the Author

Reviewer #1: I felt very excited when I received the invitation to be a reviewer of this manuscript. Despite a little bit long, the title of the paper gave me a lot of expectations about an integrative ecophysiology study that would mix a least two different levels of organization: 1. the cellular, analyzing changes in the oxidative balance, and 2. the organismal, associating thermal biology with dehydration. Unfortunately, this expectation has not been totally covered across the manuscript. In my opinion, the description of the methods and results related to the redox balance of the animals are good, despite the bad presentation of table 1 and the fact that the authors just test one antioxidative enzyme (SOD). On the other hand, where I found more weaknesses were in the link between the result of the oxidative stress of the individuals and the data of body temperature and its association with the dehydration of the animals (if exist and/or if it is significant for the oxidative stress process).

To begin, I did not find the data of body temperature that the authors reported in the section of materials and methods (measured with a temperature-sensitive PIT). These data are the backbone that supports all the thermal biology analyses, critically required to understand the impact of the incubator heat stress in the animals and the simulated heat wave experiments. Likewise, the authors do not show an analysis about the dehydration process between the treatments (and/or among the individuals of the same treatment), which is a great error because one of their main conclusion is related to a critical role of magnitude, duration and water availability for the stress experienced by each tissue and its influences in redox balance of each one of them. Further comments on the manuscript are presented in the following section. I can see great potential in this study, and I suggest the authors to organize more the result presentation to get the link between thermal biology (dehydration) and oxidative stress analyses.

General commentaries

- Line 24–26: excessive use of parenthesis () in one single phrase.

- Line 26–28: Here the main result of the manuscript: “heat wave produces brain oxidative damage and in absence of water these heat waves can damage the liver and the kidney”, but the authors do not show the information of body temperature or dehydration.

- Line 46–47. which measurements did the authors use to determine the temperatures in the local habitat of the specimen?

- line 58–59. Some Keywords are repeated in the title

- Until line 72, I did not read the connection between sublethal effects of heat waves related to oxidative damage rise. Literature review about this topic is extremely necessary.

- Lines 95–101. Although it is true that oxidative stress produces damages in several subcellular structures and disruption in some metabolic pathways, it is also true that the production of low quantities of ROS is very important triggers for strategies against oxidative damages. I strongly recommend that the authors read papers such as: DOI: 10.1016/j.freeradbiomed.2015.07.156 or DOI: 10.1016/j.cbpa.2019.04.004, and especially have a reflection about the concept of “hormeses” and its importance for the knowledge of the biochemistry ecology (related to oxidative balance) of wild animals of extreme environments (DOI: 10.3389/fphys.2018.00945). Since ~2000, to consider the ROS production as strictly “a problem” could be an idea out of date.

- Lines 104–106. Several studies evaluate measurements of oxidative stress in a variety of tissues as a consequence of exposure to different thermal regimes in numerous animals. I did not understand if the authors referred to these studies in Rhabdomys dilectus ?.

- Lines 113–123. I find unnecessary and too long justification of why the authors choose the brain, kidney, and liver as tissue models for the redox tests.

- Lines 174–179. The authors deserve a high praise for the methodology chosen for the temperature measurement. Even more by the hydration control done. However, I did not find these records in the results or discussion.

- Lines 206–208. How authors can be sure that do not exist “heat stress” in animal field conditions under the influence of a daily temperature cycle from an average summer day? I did not find any data about the heat stress of R. dilectus in the field. In my opinion, the author still can use the “Control” group as they delimitated, but they cannot assume the absence of heat stress in the field for the model species without data related.

- Line 221. Climate condition? Climate condition means all the conditions associated with environmental factors in the habitat of the species. In this experiment, the authors just controlled the temperature (it does not mean that they had little work in the design of the experiment). I felt some confusion in this section with the terms CLIMATE VS WEATHER.

- Line 235. From my point of view, here is the biggest problem in the experimental design of these experiments. It is well known that the use of volatile anesthetics, as “isoflurane”, induce oxidative stress (https://www.ncbi.nlm.nih.gov/pmc/articles/PMC4458520/), that is why I will ask the authors an explication of how can they be sure that the result obtained (by temperature treatments) are not influenced by the overdoses of isoflurane.

- Line 419. Table 1 is unnecessary. If the authors want to use a table, I believe that it is a better idea to show a summary of all the oxidative stress and oxidative damage parameters in one single table and summarizing paragraphs of the results.

• Lines 434–440. I did not find where the authors showed the result that underpins this part of the discussion (the thermal biology and water balance discussion). Mainly, I am very curious to know the data related to “heat gain and high heat loads in tissues”. Without these supports, all the discussion loses its validity.

• Lines 467–470. Until these lines, I had not seen any result about water balance in this manuscript. First of all, the percentage of body mass lost have to be in the results section, not here in the discussion. A complete water balance analysis (statistical analysis) is required for this manuscript to substance this discussion. Finally, I require the authors a better explanation about how a slight body mass lost (~2% of body mass lost) can be related to a very grave physiological state as is the hyperosmolarity (talking in terms of magnitude).

• Lines 490… Climate change, related to temperature? Climate changes it much more than global warming. I assume that the authors are talking about some “warming effect”. I suggest moderation in this paragraph, especially because the authors do not show results about behavioral or physiological thermoregulation (or thermoconformation) of the species in its microhabitats. I am not discarding the possibility, but the authors do not show data on the environmental physiology of the species to support this kind of discussion.

• Line 500: “animals from arid regions, which are already living at their physiological limit”. I strongly suggest to the authors think about this phrase. I felt it as an “anthropocentric view” totally outside of actual knowledge of the environmental physiology of animals of extreme environments. I highlight this phrase because the authors do not show data that support that their model species (or other animals from arid regions) is at the edge of their physiology. I certainly do not rule out this possibility with some species, but in the light of the actual literature of environmental physiology, it is a mistake to generalize it.

Reviewer #2: This manuscript aims to understand the role of heat stress on antioxidant defenses and oxidative damage in wild African mice. The authors had a heat stress group and a heat wave one and measured multiple markers of oxidative damage and antioxidants to determine if oxidative stress/damage occurred in response to heat. A cool design! My main concerns are the preliminary nature of the results and the treatment of dehydration in the MS. A small number of mice, changes in 3 of the 24 samples measured (4 biochem analysis/tissue-three tissues on 2 experiments), and no damage/defense consensus across heat experiments. Add to that the possibility that 2 of those 3 changes could be dehydration related (not heat), and we have essentially a very mild effect of heat stress and no other data of any type.

There is some level of recognition by the authors, albeit small, that heat stress comes with dehydration. Specifically, on the heat stress experiment, the mice were dehydrated. This is seen in the body weight loss (most likely event here is water loss), but the authors still refer to their stress as heat and talk about hyperthermia effects and heat loads. That experiment is a heat and dehydration stress experiment, and the results should also be treated as such. SOD is elevated but TAC is not, and I think that is because SOD is elevated in response to dehydration (from plants to humans). The authors should talk more about the role of dehydration in oxidative stress in their intro but also the discussion as they talk about SOD.

Another concern is the timeline of the experiments and sampling. According to the MS, the mice were euthanized immediately after the treatment, rather than offering a recovery period that would have allowed for some of the damage, defenses, or both to build up. Specifically, MDA levels responding to stress will continue to increase during recovery allowing for a more accurate measure of what happened. By cutting that time short, we are not getting the full picture and the authors have missed out on the full picture. Obviously, this cannot be done now, nor would I suggest it, but could the authors address why the preferred to do it this way, when other mammalian studies use more prolonged timelines and recovery periods.

Specific comments:

Line 52: there is acknowledgement here that water availability is an issue in heat stress. But being without water during heat stress is not a heat stress issue, it is a dehydration one. Meaning; water loss during heat might be a characteristic of heat exposure but it is still a dehydration event; the two cannot be separated in this experiment. The authors should address this.

Line 97: Some of the oxidative stress associated with heat stress is due to the increase in oxidation brought about by dehydration (which can affect metabolism just like heat exposure can). This is the perfect place to elaborate on dehydration damage, because dehydration damage and heat damage cannot be separate in the current experimental design.

Line 151: “Mice…experiment”. This information was just mentioned on the previous three sentences.

Line 171: If this is accurate and water was not provided then this is a heat plus dehydration experiment.

Line 175: while adding saline will likely buffer for osmotic issues, dehydration still occurred during the heat exposure. I would not expect saline to be able to completely prevent/overcome dehydration symptoms.

Lines 245-257: these two paragraphs are essentially the same aside form the type of homogenizer. They can be combined into one paragraph without the need for multiple “identical” sentences.

Lines 287-305: couldn’t the authors just cite the very common protein carbonyl protocol and add any modifications that they did for their species. A lot of the information here is not experiment specific and just common steps in a protocol.

Line 377: The authors should use oxidative damage to lipids rather than MDA to make it better for the reader. We are not really interested in MDA but rather what MDA represents; lipid peroxidation. Same with PC. Ox damage to proteins or damaged proteins, etc. It makes the damage seem more relevant that way.

Line 385: SOD levels were higher in liver but not TAC. I would not have expected that difference with heat stress. I wonder about dehydration though.

Line 426: This statement is not supported by the data and is a little misleading. One marker of oxidative stress increased in response to heat stress and a different marker increased in response to heat and dehydration stress. And neither one of them increased universally. These were tissue specific increases, that while very important, do not suggest thermal stress alters oxidative balance in the whole mouse.

Line 450: Another statement that does not directly flow from the data. The increase of SOD does not indicate the heat load was high. It indicates the heat and dehydration were enough to elicit that upregulation. Because dehydration cannot be separate from heat in this experiment, the authors must treat them as a combined treatment otherwise their conclusion is not scientifically sound.

Line 466: again, not heat but heat plus/and dehydration. The fact that the kidney is the only measured tissue in this experiment that had an increase in damage, suggests to me that dehydration is more damaging to kidneys than heat was. I think this is super cool! And it should be treated as much as a potential effect of heat as one from dehydration; maybe it is both! In the context of climate change there is a lot of focus on temperature. But stress is complex and responses multifarious. If the heat don’t get you, the dryness will!

Line 469: Yes! More of this please! I agree and I think the authors should focus on heat+water loss rather than make statements like the one in 466 above about hyperthermia.

Reviewer #3: PONE-D-20-14410

"Heat induced oxidative damage and antioxidant defenses following incubator heat stress and a simulated heat wave in wild caught four-striped field mice Rhabdomys dilectus

The authors report in this ms, the response to heat stress of wild caught four-striped field mice Rhabdomys dilectus by analyzing oxidative damage and antioxidant defenses. They analyzed liver, kidney and brain and quantified lipid peroxidation by MDA and protein carboylation as markers for oxidative damage and superoxide dismutase activity and total antioxidant capacity as markers for antioxidant defenses.

It is an interesting article and contribution. The methods used are clearly presented, appropriate and the results and discussion are well presented.

6. PLOS authors have the option to publish the peer review history of their article (what does this mean?). If published, this will include your full peer review and any attached files.

Reviewer #1: No

Reviewer #2: No

Reviewer #3: No

---

## [Author Response · Author response to Decision Letter 0]

11 Aug 2020

Repsonse to Editor

Corrections have been made to the journal style for the title and author affiliations as well as headings within the manuscript. Figure legends have been adjusted. Ethics statement has been added to the methods and information regarding permission to capture animals has also been added. Figures have also been adjusted and follow the necessary requirements.

Response to reviewers

The authors wish to thank each reviewer for their insightful input and comments. We also wish to thank reviewer 3 for their positive feedback. Original reviewer comments are kept green for ease of reading. Lines numbers are corresponding to the manuscript with track changes.

Reviewer #1: I felt very excited when I received the invitation to be a reviewer of this manuscript. Despite a little bit long, the title of the paper gave me a lot of expectations about an integrative ecophysiology study that would mix a least two different levels of organization: 1. the cellular, analyzing changes in the oxidative balance, and 2. the organismal, associating thermal biology with dehydration. Unfortunately, this expectation has not been totally covered across the manuscript. In my opinion, the description of the methods and results related to the redox balance of the animals are good, despite the bad presentation of table 1 and the fact that the authors just test one antioxidative enzyme (SOD). On the other hand, where I found more weaknesses were in the link between the result of the oxidative stress of the individuals and the data of body temperature and its association with the dehydration of the animals (if exist and/or if it is significant for the oxidative stress process).

To begin, I did not find the data of body temperature that the authors reported in the section of materials and methods (measured with a temperature-sensitive PIT). These data are the backbone that supports all the thermal biology analyses, critically required to understand the impact of the incubator heat stress in the animals and the simulated heat wave experiments. Likewise, the authors do not show an analysis about the dehydration process between the treatments (and/or among the individuals of the same treatment), which is a great error because one of their main conclusion is related to a critical role of magnitude, duration and water availability for the stress experienced by each tissue and its influences in redox balance of each one of them. Further comments on the manuscript are presented in the following section. I can see great potential in this study, and I suggest the authors to organize more the result presentation to get the link between thermal biology (dehydration) and oxidative stress analyses.

We thank the reviewer for their input, several changes have been made to the manuscript. Firstly, greater emphasis has been placed on the importance of dehydration to the oxidative stress process, and less so on the magnitude and duration of heat stress. Several changes to the text have been made with this in mind. Body temperatures were not recorded with PIT tags for the simulated heat wave since it was impractical for several reasons. Firstly, we were using males in this experiment and we observed in a previous experiment that the tags move in the abdomens of the animals, and end up near the testes, thus not measuring core body temperature, but several degrees lower. Tb corrections can be made for a few hours after obtaining a rectal reading, but since we were doing a behavioural experiment that lasted for several days, this was not possible. Obtaining a rectal Tb reading would also upset the animals and hence our behavioural results. Secondly, the experimental cages used during the simulated heatwave were much larger than the PIT tag reader range, and depending on where the animal was located in the cage, can result in long periods of no pit tag measurements. Due to the experimental differences in the laboratory study and the simulated heatwave a direct comparison for dehydration status between the treatments were not possible. It is therefore stated that the incubator heat stress experiment was also considered to be a dehydration experiment in this regard, but not the simulated heat wave. PIT tag Tb data, individual body mass and body mass % change data is now provided as supplementary tables, while the means are provided in the text of the manuscript.

General commentaries

- Line 24–26: excessive use of parenthesis () in one single phrase.

Lines 25-27 Changed to remove the excessive use of parenthesis ().

- Line 26–28: Here the main result of the manuscript: “heat wave produces brain oxidative damage and in absence of water these heat waves can damage the liver and the kidney”, but the authors do not show the information of body temperature or dehydration.

Body temperature data is added as supplementary material for the incubator experiment to support claim that the animals were hyperthermic (i.e. above 39oC body temperature). The body mass loss data were added as a supplementary table. Partial correlation analyses were performed to support that dehydration (due to changes in body mass before and after) occurred and is correlated to a respective oxidative marker. The degree of dehydration is supported with references along with visual inspection of the animals following the incubator experiments (they were wet from salivary spreading) (not included in text). Dehydration was unlikely to occur during the simulated heat waves as increased drinking of water occurred to maintain a hydrated state as observed in the study by Jacobs et al. (2020). 

- Line 46–47. which measurements did the authors use to determine the temperatures in the local habitat of the specimen?

Lines 49-50: Weather station data was used, this is included in the text now. 

- line 58–59. Some Keywords are repeated in the title

Lines 65-66: Some keywords were changed for synonyms.

- Until line 72, I did not read the connection between sublethal effects of heat waves related to oxidative damage rise. Literature review about this topic is extremely necessary.

This may due to a misunderstanding. Oxidative stress was not mentioned with regards to sublethal effects as a consequence of heat waves induced from climate change. It was alluded to later in the introduction that oxidative stress is just a means to measure the physiological state of animals to allude to possible sub-lethal consequences if there are any. 

- Lines 95–101. Although it is true that oxidative stress produces damages in several subcellular structures and disruption in some metabolic pathways, it is also true that the production of low quantities of ROS is very important triggers for strategies against oxidative damages. I strongly recommend that the authors read papers such as: DOI: 10.1016/j.freeradbiomed.2015.07.156 or DOI: 10.1016/j.cbpa.2019.04.004, and especially have a reflection about the concept of “hormeses” and its importance for the knowledge of the biochemistry ecology (related to oxidative balance) of wild animals of extreme environments (DOI: 10.3389/fphys.2018.00945). Since ~2000, to consider the ROS production as strictly “a problem” could be an idea out of date.

Lines 115-118: We added the relevance of ROS to the physiological response of the body under normal and under other physiological stresses.

- Lines 104–106. Several studies evaluate measurements of oxidative stress in a variety of tissues as a consequence of exposure to different thermal regimes in numerous animals. I did not understand if the authors referred to these studies in Rhabdomys dilectus ?.

Lines 132: We added the species to the sentence for clarification.

- Lines 113–123. I find unnecessary and too long justification of why the authors choose the brain, kidney, and liver as tissue models for the redox tests.

We decided to keep this in a manuscript due to other reviewers not having any objections to this being in a text and add to the literature review of how tissues can vary in their responses.

- Lines 174–179. The authors deserve a high praise for the methodology chosen for the temperature measurement. Even more by the hydration control done. However, I did not find these records in the results or discussion.

Line 214: PIT tag data added as supplementary material.

- Lines 206–208. How authors can be sure that do not exist “heat stress” in animal field conditions under the influence of a daily temperature cycle from an average summer day? I did not find any data about the heat stress of R. dilectus in the field. In my opinion, the author still can use the “Control” group as they delimitated, but they cannot assume the absence of heat stress in the field for the model species without data related.

Lines 237-238: The sentence has been rephrased to suggest minimal to no heat stress, at least when compared to the simulated heat wave treatment, due to the lack of wild thermal stress data of R. dilectus. 

- Line 221. Climate condition? Climate condition means all the conditions associated with environmental factors in the habitat of the species. In this experiment, the authors just controlled the temperature (it does not mean that they had little work in the design of the experiment). I felt some confusion in this section with the terms CLIMATE VS WEATHER.

Lines 260-261: References to climate and weather were changed to temperatures as weather or climate could not be completely recreated inside these temperature control rooms. 

- Line 235. From my point of view, here is the biggest problem in the experimental design of these experiments. It is well known that the use of volatile anesthetics, as “isoflurane”, induce oxidative stress (https://www.ncbi.nlm.nih.gov/pmc/articles/PMC4458520/), that is why I will ask the authors an explication of how can they be sure that the result obtained (by temperature treatments) are not influenced by the overdoses of isoflurane.

The paper https://www.ncbi.nlm.nih.gov/pmc/articles/PMC4458520/ explicitly uses the isoflurane for anesthetic purposes lasting over an hour and not intended to kill the animal. However, as stated this does not completely exempt the influences of isoflurane and that oxidative damage may have been underestimated in some tissues. The benefits of isoflurane however allow for the rapid initiation of tissue collection to minimise the degradation of metabolites to preserve the tissues due to several tissues being collected as stated by https://www.ncbi.nlm.nih.gov/pmc/articles/PMC4458520/. SOD activity under anaesthetized rats also did not significantly differ. Moreover, many recent studies have used isoflurane to euthanized animals in oxidative stress studies DOI: 10.4314/tjpr.v19i1.10, https://doi.org/10.3390/antiox9040332, https://doi.org/10.3892/etm.2017.5653. It was therefore expected that changes under euthanasia to oxidative stress should not significantly influence the data. In the future, alternative euthanasia methods will be used to further minimise any possible factors in oxidative stress measurements.

- Line 419. Table 1 is unnecessary. If the authors want to use a table, I believe that it is a better idea to show a summary of all the oxidative stress and oxidative damage parameters in one single table and summarizing paragraphs of the results.

Table removed.

• Lines 434–440. I did not find where the authors showed the result that underpins this part of the discussion (the thermal biology and water balance discussion). Mainly, I am very curious to know the data related to “heat gain and high heat loads in tissues”. Without these supports, all the discussion loses its validity.

Lines 119-123: Literature review on dehydration effects was added to the discussion.

Lines 477-478: Heat gain and heat loads were referenced in response to the paper cited. It is to discuss the results between the two experimental designs as tissues differed in the duration of heat exposure.

• Lines 467–470. Until these lines, I had not seen any result about water balance in this manuscript. First of all, the percentage of body mass lost have to be in the results section, not here in the discussion. A complete water balance analysis (statistical analysis) is required for this manuscript to substance this discussion. Finally, I require the authors a better explanation about how a slight body mass lost (~2% of body mass lost) can be related to a very grave physiological state as is the hyperosmolarity (talking in terms of magnitude).

Lines 420-426: Results about the incubator heat stress water balance.

Lines 444-449: Results about the simulated heat wave water balance.

Lines 528-530: This was a mistake, the 2% should have been 2g. The proper values have been used in the manuscript, with the individual data provided as supplementary material. Body mass data is tabulated from before and after with a %change in body mass which does not include the saline injection which was given to allow animals to be in a hydrated state before entering the experiment. The larger % loss supported by the literature would suggest dehydration occurred and the oxidative stress correlation along with the literature would suggest the hyperosmolality may likely have been at least one of the primary causes.

Lines 468-471: The body mass and dehydration discussion.

• Lines 490… Climate change, related to temperature? Climate changes it much more than global warming. I assume that the authors are talking about some “warming effect”. I suggest moderation in this paragraph, especially because the authors do not show results about behavioral or physiological thermoregulation (or thermoconformation) of the species in its microhabitats. I am not discarding the possibility, but the authors do not show data on the environmental physiology of the species to support this kind of discussion.

Lines 559-564: Statement changed to be more in line with the current study measurement in the absence of behavioural and physiological thermotolerance. 

• Line 500: “animals from arid regions, which are already living at their physiological limit”. I strongly suggest to the authors think about this phrase. I felt it as an “anthropocentric view” totally outside of actual knowledge of the environmental physiology of animals of extreme environments. I highlight this phrase because the authors do not show data that support that their model species (or other animals from arid regions) is at the edge of their physiology. I certainly do not rule out this possibility with some species, but in the light of the actual literature of environmental physiology, it is a mistake to generalize it.

This statement was removed to minimise speculation.

Reviewer #2: This manuscript aims to understand the role of heat stress on antioxidant defenses and oxidative damage in wild African mice. The authors had a heat stress group and a heat wave one and measured multiple markers of oxidative damage and antioxidants to determine if oxidative stress/damage occurred in response to heat. A cool design! My main concerns are the preliminary nature of the results and the treatment of dehydration in the MS. A small number of mice, changes in 3 of the 24 samples measured (4 biochem analysis/tissue-three tissues on 2 experiments), and no damage/defense consensus across heat experiments. Add to that the possibility that 2 of those 3 changes could be dehydration related (not heat), and we have essentially a very mild effect of heat stress and no other data of any type.

There is some level of recognition by the authors, albeit small, that heat stress comes with dehydration. Specifically, on the heat stress experiment, the mice were dehydrated. This is seen in the body weight loss (most likely event here is water loss), but the authors still refer to their stress as heat and talk about hyperthermia effects and heat loads. That experiment is a heat and dehydration stress experiment, and the results should also be treated as such. SOD is elevated but TAC is not, and I think that is because SOD is elevated in response to dehydration (from plants to humans). The authors should talk more about the role of dehydration in oxidative stress in their intro but also the discussion as they talk about SOD.

Another concern is the timeline of the experiments and sampling. According to the MS, the mice were euthanized immediately after the treatment, rather than offering a recovery period that would have allowed for some of the damage, defenses, or both to build up. Specifically, MDA levels responding to stress will continue to increase during recovery allowing for a more accurate measure of what happened. By cutting that time short, we are not getting the full picture and the authors have missed out on the full picture. Obviously, this cannot be done now, nor would I suggest it, but could the authors address why the preferred to do it this way, when other mammalian studies use more prolonged timelines and recovery periods.

We agree that dehydration plays a much larger role by itself and/or in conjunction with heat stress to induce oxidative stress. To emphasise - dehydration, additional statistical analyses were performed. In response to why the mice were euthanized immediately, this study was based on the protocol by Ostberg Kapla and Repasky (2002). We chose to sacrifice animals immediately after the treatment to observe the oxidative damage and antioxidant activity of individuals before any further damage or recovery occurred. As recovery might be likely post-rehydration resulting inobserved significant enzymatic activity in stressed tissues.

Specific comments:

Line 52: there is acknowledgement here that water availability is an issue in heat stress. But being without water during heat stress is not a heat stress issue, it is a dehydration one. Meaning; water loss during heat might be a characteristic of heat exposure but it is still a dehydration event; the two cannot be separated in this experiment. The authors should address this.

Several changes has been made throughout the manuscript: Emphasis has been placed on the combined effect of heat stress and dehydration. The authors agree that the presence of water is only important to circumvent dehydration and to allow for rehydration, and that dehydration is the oxidative stressor and not the availability of water.

Line 97: Some of the oxidative stress associated with heat stress is due to the increase in oxidation brought about by dehydration (which can affect metabolism just like heat exposure can). This is the perfect place to elaborate on dehydration damage, because dehydration damage and heat damage cannot be separate in the current experimental design.

Lines 119-123: Added dehydration literature review.

Line 151: “Mice…experiment”. This information was just mentioned on the previous three sentences.

Lines 160: This section was removed.

Line 171: If this is accurate and water was not provided then this is a heat plus dehydration experiment.

This has been taken into consideration and dehydration was also included in the incubator experiment. 

Line 175: while adding saline will likely buffer for osmotic issues, dehydration still occurred during the heat exposure. I would not expect saline to be able to completely prevent/overcome dehydration symptoms.

This was true as body mass was still lost in addition to the saline given, we elaborate on this later in the text.

Lines 245-257: these two paragraphs are essentially the same aside form the type of homogenizer. They can be combined into one paragraph without the need for multiple “identical” sentences.

Lines 281-291: Comment taken into consideration and changes were made to reduce the amount of text.

Lines 287-305: couldn’t the authors just cite the very common protein carbonyl protocol and add any modifications that they did for their species. A lot of the information here is not experiment specific and just common steps in a protocol.

Lines 313-330: Text removed that have similarities in the original protocol and only differences in the protocol were reported, which were 1 ml 20% TAC instead of 125 µL 50% TAC. 

Line 377: The authors should use oxidative damage to lipids rather than MDA to make it better for the reader. We are not really interested in MDA but rather what MDA represents; lipid peroxidation. Same with PC. Ox damage to proteins or damaged proteins, etc. It makes the damage seem more relevant that way.

Lines: 400-439: Relevant changes have been made and throughout the text where necessary. 

Line 385: SOD levels were higher in liver but not TAC. I would not have expected that difference with heat stress. I wonder about dehydration though.

Lines 487-493: Dehydration oxidative stress has been reviewed in the introduction and elaborated on in the discussion. 

Line 426: This statement is not supported by the data and is a little misleading. One marker of oxidative stress increased in response to heat stress and a different marker increased in response to heat and dehydration stress. And neither one of them increased universally. These were tissue specific increases, that while very important, do not suggest thermal stress alters oxidative balance in the whole mouse.

Lines 465-475: Statement changed to be more in line with current findings and dehydration. Our partial correlation analyses support your claims as dehydration did play a role during the incubator heat stress, but not necessarily a significant amount during the simulated heat wave.

Line 450: Another statement that does not directly flow from the data. The increase of SOD does not indicate the heat load was high. It indicates the heat and dehydration were enough to elicit that upregulation. Because dehydration cannot be separate from heat in this experiment, the authors must treat them as a combined treatment otherwise their conclusion is not scientifically sound.

Statement has been revised to show relevance to dehydration. 

Line 466: again, not heat but heat plus/and dehydration. The fact that the kidney is the only measured tissue in this experiment that had an increase in damage, suggests to me that dehydration is more damaging to kidneys than heat was. I think this is super cool! And it should be treated as much as a potential effect of heat as one from dehydration; maybe it is both! In the context of climate change there is a lot of focus on temperature. But stress is complex and responses multifarious. If the heat don’t get you, the dryness will!

Lines 513-541: Kidney oxidative damage discussion.

Line 469: Yes! More of this please! I agree and I think the authors should focus on heat+water loss rather than make statements like the one in 466 above about hyperthermia.

Lines 513-541: Kidney oxidative damage extention into how hyperosmolality causes oxidative damage.

---

## [Decision Letter · Decision Letter 1]

24 Sep 2020

PONE-D-20-14410R1

Heat and dehydration induced oxidative damage and antioxidant defenses following incubator heat stress and a simulated heat wave in wild caught four-striped field mice Rhabdomys dilectus

PLOS ONE

Dear Dr. Jacobs,

Thank you for submitting your manuscript to PLOS ONE. After careful consideration, we feel that it has merit but does not fully meet PLOS ONE’s publication criteria as it currently stands. Therefore, we invite you to submit a re-revised version of the manuscript that addresses the points raised during the review process.

Please, consider the last observations of referee #1.

We look forward to receiving your revised manuscript.

Kind regards,

Marcelo Hermes-Lima, PhD

Academic Editor

PLOS ONE

Reviewers' comments:

Reviewer's Responses to Questions

**Comments to the Author**

1. If the authors have adequately addressed your comments raised in a previous round of review and you feel that this manuscript is now acceptable for publication, you may indicate that here to bypass the “Comments to the Author” section, enter your conflict of interest statement in the “Confidential to Editor” section, and submit your "Accept" recommendation.

Reviewer #1: All comments have been addressed

Reviewer #2: All comments have been addressed

Reviewer #3: All comments have been addressed

2. Is the manuscript technically sound, and do the data support the conclusions?

Reviewer #1: Yes

Reviewer #2: Yes

Reviewer #3: Yes

3. Has the statistical analysis been performed appropriately and rigorously? 

Reviewer #1: Yes

Reviewer #2: Yes

Reviewer #3: Yes

4. Have the authors made all data underlying the findings in their manuscript fully available?

Reviewer #1: Yes

Reviewer #2: (No Response)

Reviewer #3: Yes

5. Is the manuscript presented in an intelligible fashion and written in standard English?

Reviewer #1: Yes

Reviewer #2: Yes

Reviewer #3: Yes

6. Review Comments to the Author

Reviewer #1: The authors accepted all the recommendations and resolved the doubts I raised in my first review. I believe that the article gained much more scientific rigor and is a good contribution to understanding the adaptive redox processes of animals that survive in extreme conditions. Nevertheless, I should like to comment one detail that do not fully satisfy me. The point is the change values of the body mass of the treatment vs the control group. Although there is a significant difference (F1 = 23.47, p = 0.001), the values present a lot of intrinsic variation in itself (differences between the control and treatments of 0.45 ± 0.66%; see S2 TABLE). Having negative treatment vs control values means that some mice were hydrated instead of dehydrated, which is not addressed by the authors (See new table S2) Outside of this points, I consider that the manuscript improved a lot from its first version.

Reviewer #2: I really appreciate that the authors took the time to address our concerns and improve their manuscript. These changes make an already good story into a solid one. I am always excited when we come together this way during the peer review process. Fantastic job!

Reviewer #3: This is an interesting work. The authors explained clearly the changes that were made and responded adequately to the reviewers.

7. PLOS authors have the option to publish the peer review history of their article (what does this mean?). If published, this will include your full peer review and any attached files.

Reviewer #1: No

Reviewer #2: No

Reviewer #3: No

---

## [Author Response · Author response to Decision Letter 1]

30 Sep 2020

We wish to thank all the reviewers for their comments and their appraisal in addressing their initial concerns and improve the manuscript.

Response to Reviewer 1:

Reviewer #1: The authors accepted all the recommendations and resolved the doubts I raised in my first review. I believe that the article gained much more scientific rigor and is a good contribution to understanding the adaptive redox processes of animals that survive in extreme conditions. Nevertheless, I should like to comment one detail that do not fully satisfy me. The point is the change values of the body mass of the treatment vs the control group. Although there is a significant difference (F1 = 23.47, p = 0.001), the values present a lot of intrinsic variation in itself (differences between the control and treatments of 0.45 ± 0.66%; see S2 TABLE). Having negative treatment vs control values means that some mice were hydrated instead of dehydrated, which is not addressed by the authors (See new table S2) Outside of this points, I consider that the manuscript improved a lot from its first version.

Lines 423-426: As the authors currently understand the control group had a positive body mass change 0.45 ± 0.66%, whereas the treatment group had a negative change. This suggests a gain in body mass (for the control group) suggesting individuals were hydrated. This can be explained as body mass measurements were taken before the saline injection. The saline injection likely increased the body mass to some extent, which was observed post-experiment. Due to the significant water losses in the treatment group, all individuals demonstrated decreased body mass changes, whereas the control group demonstrated some individuals who had a net gain in body mass.

---

## [Editor Report · Decision Letter 2]

30 Oct 2020

Heat and dehydration induced oxidative damage and antioxidant defenses following incubator heat stress and a simulated heat wave in wild caught four-striped field mice Rhabdomys dilectus

PONE-D-20-14410R2

Dear Dr. Paul Jacobs,

We’re pleased to inform you that your manuscript (revised version) has been judged scientifically suitable for publication and will be formally accepted for publication once it meets all outstanding technical requirements.

Kind regards,

Marcelo Hermes-Lima, PhD

Academic Editor

PLOS ONE
---

## [Editor Report · Acceptance letter]

4 Nov 2020

PONE-D-20-14410R2 

**Heat and dehydration induced oxidative damage and antioxidant defenses following incubator heat stress and a simulated heat wave in wild caught four-striped field mice *Rhabdomys dilectus***

Dear Dr. Jacobs:

I'm pleased to inform you that your manuscript has been deemed suitable for publication in PLOS ONE. Congratulations! Your manuscript is now with our production department. 

Kind regards, 

on behalf of

Dr. Marcelo Hermes-Lima 

Academic Editor

PLOS ONE